# ENSEMBLES AND COCKTAILS: ROBUST FINETUNING FOR NATURAL LANGUAGE GENERATION

## ABSTRACT

When finetuning a pretrained language model for natural language generation tasks, one is currently faced with a tradeoff. *Lightweight finetuning* (e.g., prefix-tuning, adapters), which freezes all or most of the parameters of the pretrained model, has been shown to achieve stronger out-of-distribution (OOD) performance than *full finetuning*, which tunes all of the parameters. However, lightweight finetuning can underperform full finetuning in-distribution (ID). In this work, we present methods to combine the benefits of full and lightweight finetuning, achieving strong performance both ID and OOD. First, we show that an ensemble of the lightweight and full finetuning models achieves the best of both worlds: performance matching the better of full and lightweight finetuning, both ID and OOD. Second, we show that we can achieve similar improvements using a single model instead of two with our proposed *cocktail finetuning*, which augments full finetuning via distillation from a lightweight model. Finally, we provide some explanatory theory in a multiclass logistic regression setting with a large number of classes, describing how distillation on ID data can transfer the OOD behavior of one model to another.

## 1 INTRODUCTION

When finetuning a pretrained language model for natural language generation tasks like summarization (Narayanan et al., 2018) and table-to-text (Gardent et al., 2017), one is currently faced with a tradeoff. One can achieve strong in-distribution (ID) performance or strong out-of-distribution (OOD) performance—but not both—by choosing one of two families of finetuning approaches. Finetuning *all* parameters of the pretrained language model achieves strong ID performance (Howard & Ruder, 2018). We call this *full finetuning*. Alternatively, freezing most of the pretrained parameters during finetuning, in, e.g., adapters (Rebuffi et al., 2017; Houlsby et al., 2019a), prefix-tuning (Li & Liang, 2021; Lester et al., 2021) and bitfit (Zaken et al., 2021), achieves stronger OOD performance than full finetuning (Li & Liang, 2021; Lester et al., 2021), but worse ID performance. We call these methods *lightweight finetuning*.

In this work, we are the first to demonstrate that this tradeoff is not necessary in natural language generation: we can achieve the best of full and lightweight finetuning both ID and OOD. We test this across three natural language generation tasks in English: summarization (Narayanan et al., 2018), table-to-text (Gardent et al., 2017), and open-domain question answering (Roberts et al., 2020; Lee et al., 2019a), and two lightweight finetuning methods: adapters and prefix-tuning. Achieving the best of both worlds is important since across our 6 settings (2 lightweight finetuning methods and 3 datasets),

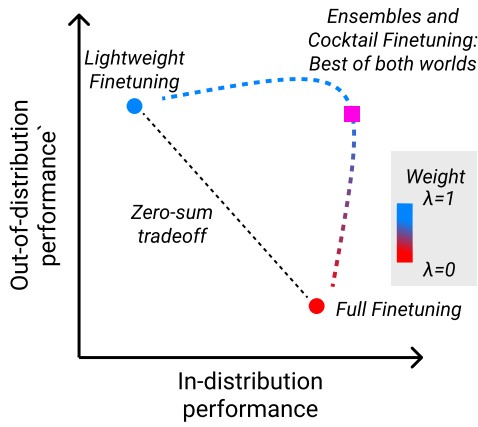

Figure 1: Instead of a zero-sum tradeoff between the ID and OOD performance of lightweight and full finetuning, ensembling and cocktail finetuning (purple square) approximately achieve the best performance of each, both ID and OOD.

full finetuning significantly outperforms lightweight finetuning ID in 5 settings, and lightweight finetuning significantly outperforms full finetuning OOD in 6 settings.

We present two methods for achieving the best of full and lightweight finetuning. We first show that a weighted ensemble of one full finetuning model and one lightweight finetuning model achieves performance comparable to the best of the two, both ID and OOD. To avoid running two models at once, we propose *cocktail finetuning*: first train a lightweight finetuning model, and then train a full finetuning model on a mixture of student-teacher distillation loss (Bucilă et al., 2006; Hinton et al., 2015) (with the lightweight model as teacher) and cross-entropy loss, on the original ID data.[1] The ensemble weight (resp. the cocktail mixture loss weight) is chosen to optimize for ID performance, and we observe that the resultant model achieves comparable OOD performance to lightweight finetuning.

Our work contributes to an ongoing realization in machine learning: robust models that deviate minimally from a pretrained model can be gainfully combined with a model adapted more precisely to a specific dataset. Most saliently, in concurrent work, Wortsman et al. (2021) find that zero-shot image recognition through CLIP (Radford et al., 2021) can be combined with a finetuned network to perform well ID and OOD. Our work demonstrates that similar results hold true in natural language generation as well. We draw from simple, well-known methods: ensembles, student-teacher distillation (Abnar et al., 2020; Hinton et al., 2015)—but our application to achieving the best of ID and OOD performance in natural language generation is novel.

To better understand how cocktail finetuning can transfer the out-of-distribution behavior of the lightweight finetuning model when distilling *only on in-distribution data*, we theoretically study multiclass logistic regression with a large number of output classes, viewing language generation as a classification problem over a large output space. We study a distribution shift setting where not all classes are seen in the ID training data (as is true in language generation due to Zipf's law (Zipf, 1949)), and getting good OOD accuracy requires generalizing to unseen outputs. First, we prove that a model trained with gradient descent cannot distinguish the unseen output classes, leading to poor OOD accuracy. Second, we prove that distilling a teacher model with good OOD accuracy on only ID training data can transfer the OOD behavior of the teacher to the student, such that the student and teacher make the same prediction on any new input. As a result, the student also has good OOD accuracy.

In summary, we present two simple methods—ensembling and cocktail finetuning—for achieving strong ID and OOD natural language generation with only the original ID data, achieving the "best of both worlds" of full and lightweight finetuning methods.

## 2 RELATED WORK

**Lightweight finetuning.** Recently, as pretrained models scale up in the number of parameters, *lightweight* finetuning techniques have been developed to avoid updating all of the parameters. These techniques update a subset of the parameters (Zaken et al., 2021), introduce a small number of new parameters between layers of the model (called "adapters") (Rebuffi et al., 2017; Houlsby et al., 2019b; Pfeiffer et al., 2020), or learn auxiliary inputs that are prepended to the models inputs (called "prompts") (Shin et al., 2020; Li & Liang, 2021; Qin & Eisner, 2021; Liu et al., 2021b; Logan IV et al., 2021; and see Liu et al., 2021a for a survey). These techniques empirically have the benefit of being able to achieve strong out-of-distribution extrapolation performance at the cost of worse in-distribution performance compared to full finetuning models both in classification and generation settings (Li & Liang, 2021; Lester et al., 2021; Zhou et al., 2021; Utama et al., 2021). In our work, we aim to overcome this trade-off by proposing techniques for achieving the best of both worlds, with a focus on language generation.

**Student-teacher distillation.** Buciluǎ et al. (2006) introduce distillation for compressing large ensembles into smaller, faster, more efficient models, and Hinton et al. (2015) generalize their method and call it knowledge distillation. Kasai et al. (2021) distill Transformers into RNNs for linear-time inference. Abnar et al. (2020) show that knowledge distillation can transfer the inductive bias of one architecture to another: for example, distilling an LSTM to a Transformer leads to a Transformer with

---

[1]We use the term *cocktail* as it is a refreshing mixture involving distillation.

representations that are more similar to independently trained LSTMs than independently trained Transformers. The efficacy of our proposed cocktail finetuning is surprising even in context of these results, as it empirically transfers just the strong OOD performance of the teacher lightweight model to the student, while *not* transferring the weak ID performance.

**Robust finetuning in vision.** In concurrent work, Wortsman et al. (2021) propose a similar approach to ours, but focus on image classification: they combine a full finetuning model with a lightweight finetuning (linear probe) model by ensembling the two models in weight space, and show that their approach maintains both good ID and OOD performance. It is yet surprising that in the structured prediction setting of natural language generation, we further show that we can achieve high OOD performance even when weight-space ensembling is impossible (for instance, since there are weights not held in common between the full finetuning model and an optimized prompt). Moreover, we further this line of work by providing a theoretical study into how distillation can transfer OOD behavior while only using ID data.

**Self-training for robustness.** In adversarial robustness, standard models achieve good accuracy on clean examples but are not robust to adversarial examples, while adversarially trained models are more robust but are less accurate on clean examples. Raghunathan et al. (2020) show that the benefits of these two models can be combined using robust self-training (RST) (Carmon et al., 2019; Uesato et al., 2019; Najafi et al., 2019) on unlabeled data to improve clean and robust accuracy. Similarly, Khani & Liang (2021) show that RST can be used to combine models with spurious features removed (which are more robust) and without removing spurious features (which are more accurate). In general OOD robustness, In-N-Out (Xie et al., 2021) uses self-training to transfer the robustness benefits of a model pre-trained with auxiliary self-supervision to a model trained with extra auxiliary features. Berthelot et al. (2021) use unlabeled data and a consistency regularization loss for domain adaptation. While many of these methods combine the benefits of two models or use a self-training/distillation loss to improve robustness, *cocktail finetuning* differs in that it performs distillation only on ID training data without any additional ID or OOD unlabeled examples.

## 3 PROBLEM STATEMENT

### 3.1 SETUP

We consider a conditional generation task from an input space $\mathcal{X}$ (e.g., a news article) to an output space $\mathcal{Y}$ (e.g., a summary of the article), where both spaces consist of sequences of tokens from a shared vocabulary $\mathcal{V}$.

**Data.** Let $P_{\text{id}}$ and $P_{\text{ood}}$ denote the distribution of $(x, y)$ pairs in-distribution and out-of-distribution, respectively, where $x \in \mathcal{X}$ and $y \in \mathcal{Y}$. In summarization, $P_{\text{id}}$ may consist of articles about world and business news while $P_{\text{ood}}$ consists of health and technology news. The training data $D_{\text{train}}$ and validation data $D_{\text{val}}$ consist of $n$ and $n_{\text{val}}$ in-distribution $(x, y)$ pairs respectively, drawn from $P_{\text{id}}$.

**Model and metrics.** Let $q_\gamma(y \mid x) = \prod_{j=1}^{\text{len}(y)} q_\gamma(y_j \mid x, y_{<j})$ be a pretrained neural autoregressive sequence model with parameters $\gamma$. Finetuning (full or lightweight) produces a finetuned model $p_\theta(y \mid x)$ from $q_\gamma$, where $\theta$ denotes the set of trainable parameters. In full finetuning, $\theta$ is the same set of parameters as $\gamma$; in lightweight finetuning, $\theta$ often refers to the parameters of inserted modules (e.g., adapters modules or prefixes). The final predictor is $f_\theta(x)$ which approximates $\arg\max_y p_\theta(y \mid x)$ through, e.g., beam search. The goal is to learn parameters $\theta$ such that the predictor $f_\theta$ obtains good ID and OOD performance:

$$S_{\text{id}}(f_\theta) = \mathbb{E}_{x,y \sim P_{\text{id}}}[\text{score}(y, f_\theta(x))] \tag{1}$$
$$S_{\text{ood}}(f_\theta) = \mathbb{E}_{x,y \sim P_{\text{ood}}}[\text{score}(y, f_\theta(x))] \tag{2}$$

where score may be defined differently for each task (e.g., exact match score or BLEU (Papineni et al., 2002b)).

| WebNLG ID ($x$) | (103 Colmore Row \| architect \| John Madin), (John Madin \| birthPlace \| Birmingham) (Birmingham \| leaderName \| Andrew Mitchell) |
|---|---|
| WebNLG ID ($y$) | John Madin was born in Birmingham (with Andrew Mitchell as a key leader) and became an architect, designing 103 Colmore Row. |
| WebNLG OOD ($x$) | [Albennie Jones \| genre \| Rhythm and blues] [Albennie Jones \| birthPlace \| Errata, Mississippi] [Rhythm and blues \| derivative \| Disco] |
| WebNLG OOD ($y$) | Albennie Jones, born in Errata, Mississippi, is a performer of rhythm and blues, of which disco is a derivative. |
| XSUM ID ($x$) | The country's consumer watchdog has taken Apple to court for false advertising because the tablet computer does not work on Australia's 4G network.Apple's lawyers said they were willing to publish a clarification.However the company does not accept that it misled customers.The Australian Competition and Consumer Commission (ACCC) said on Tuesday: [... 170 words] corrective advertising and refunds to consumers. On its website, Apple does state that 4G LTE is only supported on selected networks in the US and Canada. |
| XSUM ID ($y$) | US technology firm Apple has offered to refund Australian customers who felt misled about the 4G capabilities of the new iPad. |
| XSUM OOD ($x$) | The 200ft (65m) structure, boasting panoramic views of London, is part of a £260m revamp of the world-famous art museum.It is being billed as the UK's most important new cultural building since the British Library.More than half of the solo displays are dedicated to women artists.At Tuesday's launch event, [... 627 words] Later this year Tate will launch Tate Exchange, an "open experiment" occupying an entire floor of the new Switch House building, that will enable 50 invited organisations from the across the UK to display their work. |
| XSUM OOD ($y$) | Tate Modern has unveiled its new extension, a pyramid-like tower housing cavernous gallery spaces, ahead of its official opening. |
| OpenQA ID ($x$) | medical term for the cause of a disease |
| OpenQA ID ($y$) | etiology |
| OpenQA OOD ($x$) | who is the newly elected governor of california? |
| OpenQA OOD ($y$) | Jerry Brown |

Table 1: Examples of in-distribution and out-of-distribution inputs and outputs from the WebNLG, XSUM and OpenQA tasks, respectively. In WebNLG, the distribution shift is due to non-overlapping topics; in XSUM, the distribution shift is due to non-overlapping news categories; in OpenQA the distribution shift is due to different data collection processes.

## 3.2 BASELINE FINE-TUNING METHODS

**Full finetuning** has been the defacto approach to adapt pretrained language models for downstream generation tasks (Lewis et al., 2020; Kale & Rastogi, 2020). Under this framework, the model $p_\theta$ is initialized with its pretrained parameters $\gamma$, and we optimize the following log-likelihood objective.

$$L(\theta) = \mathbb{E}_{(x,y) \sim D_{\text{train}}}[-\log p_\theta(y \mid x)] \tag{3}$$

where the set of trainable parameters $\theta$ refer to the set of pretrained parameters $\gamma$.

**Lightweight finetuning** is another family of adaptation approaches, which freezes most of the pretrained parameters and augments the model with small trainable modules. Lightweight finetuning shares the same training objective as full finetuning, but the trainable parameters $\theta$ are different. Within the lightweight finetuning family, we focus on adapter-tuning (Houlsby et al., 2019a) and prefix-tuning (Li & Liang, 2021). Both approaches keep the pretrained parameters intact. Adapter-tuning inserts task-specific MLP layers between each layer of the pretrained language models, and the trainable parameter $\theta$ consists of these inserted MLP parameters. Prefix-tuning prepends a sequence of trainable, task-specific prefix vectors to the input, and $\theta$ consists of these prefix parameters.[2]

## 4 EXPERIMENT SETUP

### 4.1 DATASETS AND METRICS

**Table-to-text.** We evaluate on the WebNLG dataset (Gardent et al., 2017), where the input $x$ is a sequence of (entity, predicate, entity) triples and the output $y$ is a natural language description that covers the input information. Each example in this dataset is labeled with one of 14 topic (e.g., airports, comic characters). We select 9 topics as in-distribution and the remaining 5 topics as out-of-distribution. We evaluate the performance using the official evaluation scripts and report the BLEU scores (Papineni et al., 2002a).

**Summarization.** We use the XSUM (Narayan et al., 2018) dataset, which is an abstractive summarization dataset on BBC news articles. Each example is labeled with a news category. We select world, UK and business categories to be in-distribution and let the remaining news categories such as health and technology to be out-of-distribution. We report the ROUGE-2 scores (Lin, 2004).

**Open-domain, closed-book QA (Open-QA).** We use the open-domain variant (Lee et al., 2019b) of the Natural Questions dataset (Kwiatkowski et al., 2019) as the in-distribution data and the Web Questions dataset (Berant et al., 2013) as the out-of-distribution data. The distribution shift is induced by different data collection process and their year of collection (roughly 2018 versus 2013). We

---

[2]For specifics of these methods, see their respective papers.

report the exact match accuracy. Note that this task, unlike most QA (e.g., SQuAD (Rajpurkar et al., 2016)), does not provide systems with a text from which to extract answers. Roberts et al. (2020) demonstrated that one could finetune an LM to generate answers from knowledge stored during pretraining.

## 4.2 Model Architectures and Hyperparameters

We use GPT-2-medium (Radford et al., 2019) as the pretrained model for table-to-text and open-domain QA tasks. GPT-2 is an autoregressive language model; therefore, we parametrize $p_\theta(y \mid x)$ by concatenating $x$, a separator token, and $y$ (e.g., [x; SEP; y]). We define the loss function to sum over the token losses that correspond to $y$.

We use BART-large (Lewis et al., 2020) as the pretrained model for the summarization task. BART uses an encoder-decoder architecture and parametrizes $p_\theta(y \mid x)$ by first feeding $x$ to the encoder, and decoding $y$ conditioned on the encoder representation.

For open-domain closed-book QA, this is to our knowledge the first time GPT-2 has been finetuned for the task. We used separate development OOD data to test a handful ($<10$) of hyperparameter settings to determine whether we'd observe a similar ID/OOD tradeoff to the other tasks. We note this because, in general, one cannot do hyperparameter optimization on OOD data; we did this to provide another testbed for our methods.

Our implementation is based on the Hugging Face Transformers package (Wolf et al., 2020). We use most of the hyperparameters suggested by the Hugging Face default setup. The hyperparameters we tune include the number of epochs, batch size, learning rate, and prefix length (for prefix-tuning), all reported in Table 3 in the appendix.

## 5 Results of full and lightweight finetuning

There is a trade-off in ID/OOD performance when choosing between full finetuning and lightweight finetuning: full finetuning consistently outperforms in-distribution, whereas lightweight finetuning consistently outperforms out-of-distribution (Li & Liang, 2021; Lester et al., 2021). To quantify this trade-off, we examine the generation performance of both methods on ID and OOD data.

As shown in Table 2, when evaluating in-distribution, full finetuning achieves significantly stronger results than lightweight finetuning in five out of six settings (across 3 datasets and two lightweight finetuning methods. Only prefix-tuning on WebNLG achieves comparable accuracy to full finetuning. When evaluating out-of-distribution, lightweight finetuning achieves significantly stronger results than full finetuning in five of six settings. Only adapter-tuning on XSUM fails to outperform full finetuning.[3]

## 6 Combining models with a Weighted Ensemble

As we observe in Section 5, full finetuning models attain better ID performance but worse OOD performance, whereas lightweight finetuning models attain better OOD performance at the cost of good ID performance. Is it possible to achieve both good ID and OOD performance simultaneously? In this section, we show that a simple ensemble of the two methods accomplishes this.

First, we train a lightweight finetuning model $p^{(\ell)}$ and a full finetuning model $p^{(f)}$ on $D_{\text{train}}$. At decoding time, we combine the prediction of the $p^{(\ell)}$ and $p^{(f)}$ by interpolating them at each token.

$$p^{(\text{Ens})}(y_t \mid x, y_{<t}) = \lambda \cdot p^{(\ell)}(y_t \mid x, y_{<t}) + (1 - \lambda) \cdot p^{(f)}(y_t \mid x, y_{<t}) \qquad (4)$$

At each time step $t$, both models output a next token distribution conditioned on the common history $y_{<t}$, and the two distributions are interpolated with a mixture weight $\lambda$. The output is generated by beam searching the ensemble distribution $p^{(\text{Ens})}(y_t \mid x, y_{<t})$.

---

[3]Though it is worth noting again that for open-domain QA, we performed a small amount of manual, exploratory hyperparameter optimization to optimize the ID of full finetuning and the OOD of lightweight finetuning.

| | WebNLG (BLEU) | | XSUM (ROUGE-2) | | OpenQA (EM) | |
| | ID | OOD | ID | OOD | ID | OOD |
|---|---|---|---|---|---|---|
| | *Prefix-Tuning* | | | | | |
| Full fine-tuning (FT) | $63.25 \pm 0.42$ | $30.39 \pm 0.64$ | $21.22 \pm 0.11$ | $15.47 \pm 0.09$ | $16.8 \pm 0.1$ | $9.8 \pm 0.2$ |
| Prefix | $63.18 \pm 0.37$ | $43.75 \pm 0.33$ | $19.75 \pm 0.07$ | $16.27 \pm 0.03$ | $8.2 \pm 0.4$ | $11.0 \pm 0.1$ |
| Ensemble (Prefix) | $65.04 \pm 0.26$ | $44.80 \pm 0.37$ | $21.99 \pm 0.04$ | $16.50 \pm 0.03$ | $17.2 \pm 0.3$ | $11.4 \pm 0.1$ |
| Cocktail (Prefix) | $65.35 \pm 0.31$ | $43.96 \pm 0.53$ | $21.80 \pm 0.20$ | $16.20 \pm 0.09$ | $17.8 \pm 0.4$ | $10.6 \pm 0.2$ |
| | *Adapter-Tuning* | | | | | |
| Full fine-tuning (FT) | $63.25 \pm 0.42$ | $30.39 \pm 0.64$ | $21.22 \pm 0.11$ | $15.47 \pm 0.09$ | $16.8 \pm 0.1$ | $9.8 \pm 0.2$ |
| Adapters | $60.43 \pm 0.24$ | $48.04 \pm 0.45$ | $19.04 \pm 1.06$ | $15.63 \pm 0.59$ | $9.7 \pm 0.4$ | $11.1 \pm 0.4$ |
| Ensemble (Adapters) | $64.57 \pm 0.17$ | $46.78 \pm 0.13$ | $21.72 \pm 0.35$ | $16.54 \pm 0.19$ | $17.3 \pm 0.2$ | $11.4 \pm 0.2$ |
| Cocktail (Adapters) | $64.61 \pm 0.17$ | $44.91 \pm 0.18$ | $21.08 \pm 0.19$ | $15.73 \pm 0.08$ | $17.8 \pm 0.2$ | $10.6 \pm 0.2$ |

Table 2: Performance of full finetuning, lightweight finetuning, our ensemble, and our cocktail finetuning model, for prefix-tuning (first half) and for adapter-tuning (second half). Cells are highlighted blue if they are statistically significantly better than lightweight finetuning (for ID columns) or full finetuning (for OOD columns) and red otherwise.

### 6.1 Choosing the Mixture Weight $\lambda$

We start with a set of $\{\lambda_i\}_{i=1}^K$ where $\lambda_i \in [0,1]$. For example we use $\lambda \in \{0., 0.1, 0.25, 0.5, 0.75, 0.9, 1.0\}$ in this experiment. We choose the mixture weight $\lambda_i$ that attains the best score on $D_{\text{val}}$.

This method of choosing $\lambda$ is useful because (1) it crucially does not use OOD data, which is typically unaccessible at model selection time, and (2) it guarantees that ID performance is not sacrificed, because we can match ID performance by setting $\lambda = 0$.

### 6.2 Results

As shown in Table 2, the ensemble matches and slightly outperforms both the ID performance skyline (achieved by full finetuning) and the OOD performance skyline (achieved by lightweight finetuning). Across all datasets, ensembling achieves significantly improved performance over full finetuning OOD, and lightweight finetuning ID. Further, in eleven out of twelve settings (across 3 datasets, 2 methods, and both ID and OOD), ensembling does not perform significantly worse than the *better* of full and lightweight finetuning.

### 6.3 Runtime and Memory

Despite its ability to combine the best of both worlds, ensembling is expensive at inference time because two models are run simultaneously, doubling the amount of GPU memory required. Furthermore, two forward passes are required, doubling the runtime for decoding.[4] For deployment efficiency, it's better to have a single model at inference time.

## 7 Cocktail Finetuning: Combining with Distillation

The goal of cocktail finetuning is to approximate the ensemble in Section 6 with a single model.[5]

### 7.1 Method

Train a lightweight finetuning model $p^{(\ell)}$. Fix mixture weight $\lambda \in [0,1]$. Train all parameters of the pretrained model $q_\gamma(y \mid x)$ on the following objective. First, define a token-level student-teacher

---

[4]The two forward passes are parallelizable, but at the cost of another 2x GPU memory.

[5]One might wish to approximate the two with a single lightweight model, a direction we explored but eventually discarded due to the difficulty of approximating a full finetuning model with a lightweight finetuning model. Anecdotally, this is evidence towards the speculation of Lester et al. (2021), that lightweight finetuning is limited in its ability to alter a pretrained LM.

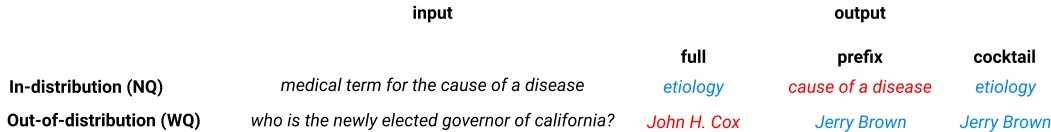

| | input | output | | |
|---|---|---|---|---|
| | | **full** | **prefix** | **cocktail** |
| **In-distribution (NQ)** | *medical term for the cause of a disease* | *etiology* | *cause of a disease* | *etiology* |
| **Out-of-distribution (WQ)** | *who is the newly elected governor of california?* | *John H. Cox* | *Jerry Brown* | *Jerry Brown* |

Figure 2: Examples in the open-domain closed-book QA (wherein no text is given from which to extract the answer). The full finetuning model performs well in-distribution but not out-of-distribution, the lightweight finetuning model performs well out-of-distribution but not in-distribution, and the cocktail finetuning model performs well on both. The distribution shift (Natural Questions to WebQuestions) is due to temporal and data collection differences.

distillation loss at token $j$:

$$\text{distill}_{p^{(\ell)}}(x, y, j) = \text{KL}\left(p^{(\ell)}(y_j \mid x, y_{<j}) \| p_\theta(y_j \mid x, y_{<j}))\right) \tag{5}$$

The objective is the sum over the length of $y$ of the $\lambda$-weighted mixture of distillation and log-likelihood loss:

$$L_\lambda(\theta) = \mathbb{E}_{(x,y) \sim D_{\text{train}}}\left[\sum_{j=1}^{\text{len}(y)} \lambda \text{distill}_{p^{(\ell)}}(x, y, j) - (1 - \lambda) \log p_\theta(y_j \mid x, y_{<j})\right] \tag{6}$$

Let $f_{\theta,\lambda}$ be the predictor resulting from optimizing on $L_\lambda(\theta)$ and estimate the ID score $S_{\text{id}}(f_{\theta,\lambda})$ on $D_{\text{val}}$. Perform this across a set of mixture weights $\{\lambda_i\}_i$. The cocktail finetuning model is the predictor $f_{\theta,\lambda_i}$ that maximizes the in-distribution score $S_{\text{id}}(f_\theta, \lambda_i)$ across the mixture weights $\lambda_i$.[6]

### 7.2 RESULTS

As shown in Table 2, cocktail finetuning matches and slightly outperforms the ID performance skyline (achieved by full finetuning) and achieves comparable to or slightly worse than the OOD performance skyline (achieved by lightweight finetuning). Across all datasets, cocktail finetuning achieves significantly improved performance over full finetuning OOD, and over lightweight finetuning ID. Further, in seven out of twelve settings (across 3 datasets, 2 methods, and both ID and OOD), cocktail finetuning does not perform significantly worse than the *better* of full and lightweight finetuning.

## 8 ANALYSIS OF THE WEIGHTING OF FULL AND LIGHTWEIGHT FINETUNING

**Trade-off curves.** Our ensembling and cocktail finetuning methods are useful in practice because one can choose the mixture coefficient $\lambda$ without OOD data. Now, we explore how ID and OOD performance vary as a function of $\lambda$. Figure 3 shows results across five random seeds of ensembling and cocktail finetuning on our three datasets, each dot colored according to its $\lambda$ value. In each plot, we see a smooth tradeoff curve, in which a range of $\lambda$ improve on the OOD, ID, or both of each constituent model. Interestingly, in two of three tasks, there is effectively only one pareto-optimal $\lambda$. This means the $\lambda$ is *best* for both for ID and OOD performance, which is why optimizing for ID performance also leads to good OOD performance.

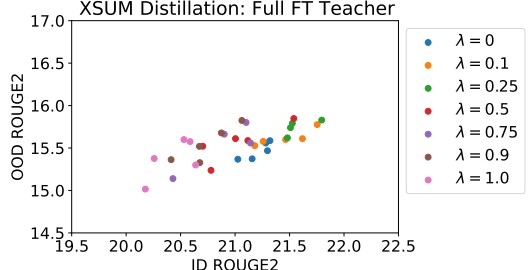

Figure 4: Ablation on XSUM in which a full finetuning model is distilled into a full finetuning model. No benefits are observed.

---

[6]For cocktail finetuning, we use the hyperparameters reported for full finetuning; no hyperparameter optimization was performed.

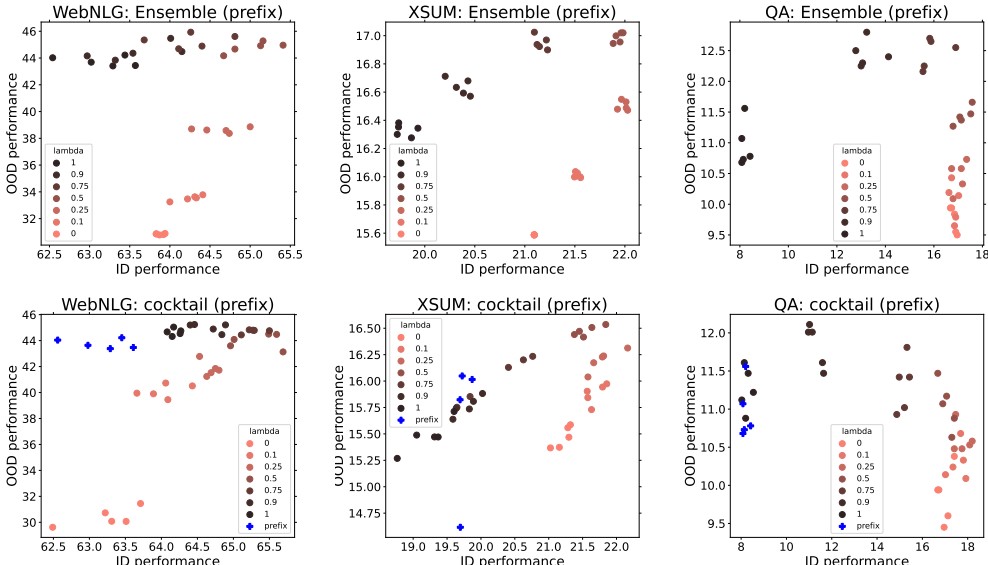

Figure 3: For WebNLG and XSUM, ensembles and cocktails both achieve the best of both prefix-tuning and full finetuning for some $\lambda$; for OpenQA, one can achieve approximately the best of both.

**Ablations.** One hypothesis to explain the strong performance of ensembles is the known benefits of ensembling two models trained with different random seeds. Likewise for cocktail finetuning, one could point to the effectiveness of born-again networks (Furlanello et al., 2018): distilling a teacher model to an identically parameterized student with the same model architecture. As an ablation to test this hypothesis, we distill from a full finetuning model into another full finetuning model, again using $\lambda$ to mix the distillation loss with the label loss. The results, in Figure 4, show that this distillation does not improve over a single full finetuning model, for any of the $\lambda$.

# 9    WHY DOES DISTILLATION ON ID DATA IMPROVE OOD?

Prior works have used unlabeled OOD data and self-training (similar to distillation) to transfer good OOD performance from another model (Raghunathan et al., 2020; Xie et al., 2021). In contrast, cocktail finetuning performs distillation only on ID data. In this section, we provide an explanation for why this can occur in multiclass logistic regression with a large number of classes.

**Prediction problem and distribution shift.** Motivated by the large output space in language generation, we consider a prediction problem from inputs $\mathcal{X} \subseteq \mathbb{R}^d$ to labels $\mathcal{Y} = \{1, \ldots, k\}$ where $k$ is large. The labeled training data is $\{(x_i, y_i)\}_{i=1}^n$. For convenience, we also let $y_i$ be the one-hot vector of length $k$ denoting the correct class. The labels in the training data do not cover all the possible labels, and we let the subset of seen labels be $\mathcal{S}$.

In this problem, the class marginals of the ID and OOD distributions $P_{\text{id}}$ and $P_{\text{ood}}$ are different. To have good OOD accuracy, the model must generalize to unseen classes.

**Model and losses.** We consider training a multi-class logistic regression model using full batch gradient descent where the learned parameters are $\theta \in \mathbb{R}^{k \times d}$. In full fine-tuning, we run gradient descent on the negative log-likelihood.

In cocktail finetuning, we also add a distillation loss using soft labels (a probability vector) from the prefix-tuned model, which we denote $p_{\tilde{\theta}}^x = p_{\tilde{\theta}}(\cdot \mid x) \in \mathbb{R}^k$ with parameters $\tilde{\theta}$, given an input $x$. Similarly, the predicted class probability vector of the cocktail model given an input $x$ is denoted

$p_\theta^x = p_\theta(\cdot \mid x) \in \mathbb{R}^k$. In this setting, the cocktail loss (Equation 6) with parameter $\lambda \in [0, 1]$ is

$$\ell_{\text{ct}}(\theta) = \sum_{i=1}^{n}((1 - \lambda)y_i + \lambda p_{\tilde{\theta}}^{x_i})^\top \log p_\theta^{x_i}. \tag{7}$$

The cocktail loss encourages the model to learn to output a mixture of the hard label and the soft label on the training data. The final score for the classification task is computed with respect to the accuracy metric, where $\text{score}(y, \hat{y}) = 1 - \ell_{0-1}(y, \hat{y})$ and $\ell_{0-1}$ is the 0-1 loss.

**Poor OOD performance with full finetuning.** Our first result shows that when training a logistic regression model using gradient descent (initialized at 0) on the only labeled training set (as in full finetuning), the learned parameters $\theta_j$ for an unseen label $j \notin \mathcal{S}$ are all the same.

**Proposition 1.** *Let $\theta^{(t)}$ be the parameters at iteration $t$ of gradient descent and let $\theta^{(0)} = 0$. At any iteration $t$ and for any unseen label index $j \notin \mathcal{S}$, $\theta_j = g_t$ where $g_t$ is the same for all $j \notin \mathcal{S}$.*

As a result, the model cannot distinguish between any of the unseen classes, resulting in poor OOD performance. Suppose that the model outputs a fixed, arbitrary unseen class $j \in \mathcal{S}$ when the probabilities of the unseen classes are the largest. Then, assuming that the OOD distribution $P_{\text{ood}}$ is such that the probability of drawing an example from an unseen class (excluding $j$) $P_{\text{ood}}(y \notin \mathcal{S} \cup \{j\})$ is at least $c \in [0, 1]$, then the OOD accuracy is bounded as

$$S_{\text{ood}} \leq 1 - P_{\text{ood}}(y \notin \mathcal{S} \cup \{j\}) \leq 1 - c \tag{8}$$

This OOD accuracy can be low when $c$ is large, which can occur with very large output spaces.

**Disillation transfers OOD behavior through soft logits.** We investigate how distillation on ID training data can affect OOD behavior. In the following, let $X \in \mathbb{R}^{n \times d}$ be the training data matrix.

**Proposition 2.** *Assume the data matrix $X$ has full column rank. Let $\theta$ be the minimizer of the cocktail loss when $\lambda = 1$, subject to the constraint that $\mathbf{1}^\top \exp(\theta X) = \alpha \mathbf{1}^\top \exp(\tilde{\theta} X)$ for some constant $\alpha > 0$, such that the model learns the same normalizing constants (up to scaling) as the prefix-tuned model on the training data only. Then*

$$p_\theta(\cdot \mid x) = p_{\tilde{\theta}}(\cdot \mid x) \tag{9}$$

*for any new input $x$, such that the cocktail model has the same OOD accuracy as the prefix model.*

Therefore, if the prefix-tuning model has high OOD accuracy, then $S_{\text{ood}}$ is high. The proof proceeds by showing that the extra information in the soft logits can help the model recover the parameters of the prefix-tuned model (up to scaling), which in turn inherits its OOD behavior. Thus, in generation problems where the output space is large, distillation on soft logits can transfer information about OOD behavior even through ID data.

## 10 CONCLUSION

Considered together, our work and the work of Wortsman et al. (2021) provide strong cross-modal evidence that simple ensembles of models that have minimally deviated from pretraining (for them, zero-shot; for us, lightweight finetuning models) with full finetuning models provides strong ID and OOD accuracy. The reasons for this "best of both worlds" behavior are still poorly understood, motivating future work. Since ensemble models effectively weight by model confidence (though modulated by $\lambda$), one explanation may be that lightweight finetuning models are less confident than full finetuning models in-distribution, yet more confident out-of-distribution.

In summary, we provide simple, effective methods for achieving the best of lightweight and full finetuning—strong OOD and ID performance—in language generation. Our cocktail finetuning provides system builders with the opportunity to optimize for in-distribution performance (in choosing lambda), often beating full finetuning, while resting assured that the out-of-distribution performance (i.e., what they observe with real users) will likely be higher than had they performed full finetuning.

ETHICS STATEMENT

The methods presented in this work have the potential to improve a wide range of natural language generation tasks. Natural language generation has known dual-use issues, like scaling misinformation generation and generating naturalistic text in scams. The WebNLG XSUM, Natural Questions, and WebQuestions datasets we evaluate on contain personal information (though public and published) information about real people and organizations.

REPRODUCIBILITY STATEMENT

All code (including experiment scripts and plot creation scripts) will be released upon publication. All datasets used in this paper are publicly available and will be linked in our released code. All reported numbers represent the average of results from 5 random seeds.

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

| | WebNLG (BLEU) | | | | XSUM (ROUGE-2) | | | | OpenQA (EM) | | | |
|---|---|---|---|---|---|---|---|---|---|---|---|---|
| | epoch | lr | batch size | other | epoch | lr | batch size | other | epoch | lr | batch size | other |
| Full fine-tuning (FT) | 5 | $5*10^{-5}$ | 6 | - | 5 | $5*10^{-5}$ | 5 | - | 6 | $5*10^{-5}$ | 16 | - |
| Adapters | 5 | $5*10^{-5}$ | 5 | d=200 | 5 | $8*10^{-5}$ | 20 | d=200 | 2 | $5*10^{-4}$ | 16 | d=200 |
| Prefix | 5 | $5*10^{-5}$ | 5 | l=20 | 15 | $5*10^{-5}$ | 64 | l=40 | 2 | $5*10^{-5}$ | 16 | l=20 |

Table 3: Hyperparameters. $d$ refers to the dimension of the adapter MLP's middle layer, and $l$ refers to the length of the prefix vector in prefix-tuning.

| | WebNLG (BLEU) | XSUM (ROUGE-2) | OpenQA (EM) |
|---|---|---|---|
| Ensemble (Adapters) | 0.5 | 0.5 | 0.5 |
| Ensemble (Prefix) | 0.5 | 0.25 | 0.5 |
| Cocktail (Adapters) | 0.5 | 0.25 | 0.25 |
| Cocktail (Prefix) | 0.5 | 0.25 | 0.25 |

Table 4: The selected values of $\lambda$ for the ensembles and cocktail experiments.

Thomas Wolf, Lysandre Debut, Victor Sanh, Julien Chaumond, Clement Delangue, Anthony Moi, Pierric Cistac, Tim Rault, Rémi Louf, Morgan Funtowicz, Joe Davison, Sam Shleifer, Patrick von Platen, Clara Ma, Yacine Jernite, Julien Plu, Canwen Xu, Teven Le Scao, Sylvain Gugger, Mariama Drame, Quentin Lhoest, and Alexander M. Rush. Transformers: State-of-the-art natural language processing. In *Proceedings of the 2020 Conference on Empirical Methods in Natural Language Processing: System Demonstrations*, pp. 38–45, Online, October 2020. Association for Computational Linguistics. URL https://www.aclweb.org/anthology/2020.emnlp-demos.6.

Mitchell Wortsman, Gabriel Ilharco, Mike Li, Jong Wook Kim, Hannaneh Hajishirzi, Ali Farhadi, Hongseok Namkoong, and Ludwig Schmidt. Robust fine-tuning of zero-shot models. *arXiv preprint arXiv:2109.01903*, 2021.

Sang Michael Xie, Ananya Kumar, Robert Jones, Fereshte Khani, Tengyu Ma, and Percy Liang. In-N-out: Pre-training and self-training using auxiliary information for out-of-distribution robustness. In *International Conference on Learning Representations (ICLR)*, 2021.

Elad Ben Zaken, Shauli Ravfogel, and Yoav Goldberg. Bitfit: Simple parameter-efficient fine-tuning for transformer-based masked language-models. *CoRR*, abs/2106.10199, 2021. URL https://arxiv.org/abs/2106.10199.

Kaiyang Zhou, Jingkang Yang, Chen Change Loy, and Ziwei Liu. Learning to prompt for vision-language models, 2021.

George Kingsley Zipf. Human behavior and the principle of least effort. 1949.

## A  Hyperparameters and Dataset Details

| | training | val | test (id) | test (ood) |
|---|---|---|---|---|
| WebNLG | 18K | 2.2K | 1.0K | 0.9 K |
| XSUM | 129K | 7k | 7k | 20k |
| OpenQA | 88K | 1.8K | 1.8K | 2k |

Table 5: Number of examples in each dataset.

### A.1  Hyperparameters

Relevant hyperparameters beyond the defaults of Huggingface Transformers Wolf et al. (2020) are reported in Table 3.

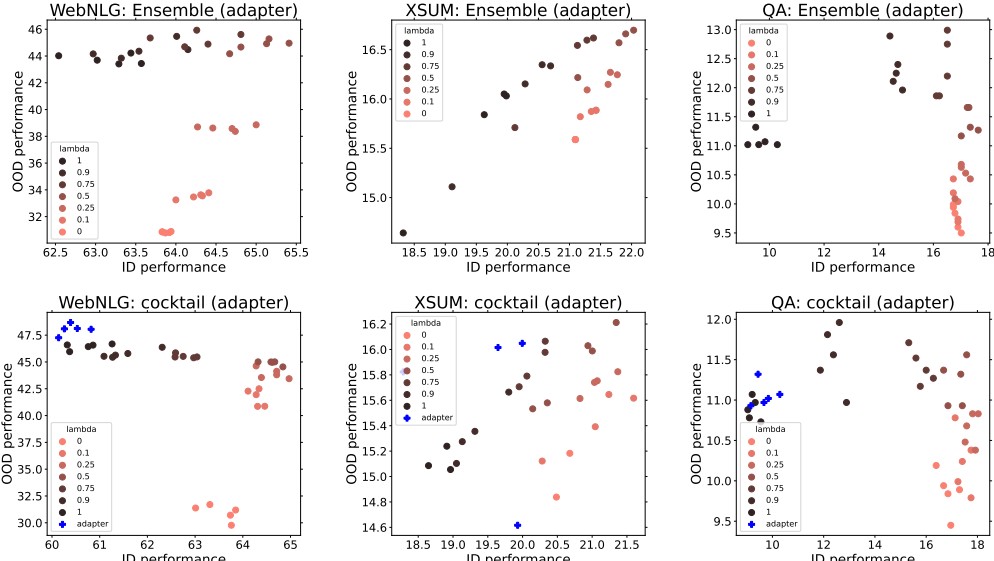

Figure 5: For WebNLG and XSUM, Ensembles and cocktails both achieve the best both prefix and full finetuning for some $\lambda$; for OpenQA, one can achieve approximately the best of both.

## A.2 DATASET DETAILS

Table 5 shows the number of examples in each dataset split.

For the WebNLG dataset, the ID topics include {Astronaut, University, Monument, Building, ComicsCharacter, Food, Airport, SportsTeam, City, and WrittenWork} and the OOD topics include {Athlete, Artist, MeanOfTransportation, CelestialBody, Politician}.

For the XSUM dataset, the ID topics include {uk, business, world } news and the OOD topics includes { entertainment, election, technology, science, health, education, explainers, live, uk-politics, world-us-canada, world-europe, uk-england, uk-scotland, world-asia, science-environment, uk-scotland-scotland-politics, disability}.

## B ADDITIONAL RESULTS

Due to space constraints, we did not include in Figure 3 the tradeoff plots using adapters lightweight finetuning method (instead reporting only those for prefix-tuning.) The plots for adapters are provided in Figure 5.

## C ANALYSIS

We restate and expand upon some of the setup for the analysis here.

**Prediction problem.** We consider a prediction problem from inputs $\mathcal{X} \subseteq \mathbb{R}^d$ to labels $\mathcal{Y} = \{1, \ldots, k\}$ where $k$ is large. The labeled training data is $\{(x_i, y_i)\}_{i=1}^n$. For convenience, we also let $y_i$ be the one-hot vector of length $k$ denoting the correct class, and will make this clear. The labels in the training data do not cover all the possible labels, and we let the subset of seen labels be $\mathcal{S}$.

**Model.** We consider training a multi-class logistic regression model using full batch gradient descent. The learned parameters are $\theta \in \mathbb{R}^{k \times d}$ and the model prediction given input $x$ is

$$f_\theta(x) = \arg\max_j p_\theta(x)_j = \arg\max_j \frac{\theta_j^\top x}{\sum_{j'} \theta_{j'}^\top x} \tag{10}$$

where $\theta_j$ is the $j$-th row of $\theta$. In standard fine-tuning, we run gradient descent on the negative log-likelihood

$$\ell(A) = -\sum_{i=1}^{n} y_i^\top \log p_\theta^{x_i} \tag{11}$$

where $y_i \in \mathbb{R}^k$ is a one-hot vector of the class label.

**Distillation loss.** In the Cocktail model, we also add a distillation loss using soft labels from the prefix-tuned model, which we denote $p_{\tilde{\theta}}$ with parameters $\tilde{\theta}$. The Cocktail loss with parameter $\lambda \in [0,1]$ is

$$\ell_{ct}(A) = \sum_{i=1}^{n} \left( (1-\lambda) y_i^\top \log p_\theta^{x_i} + \lambda (p_{\tilde{\theta}}^{x_i})^\top \log p_\theta^{x_i} \right) \tag{12}$$

$$= \sum_{i=1}^{n} ((1-\lambda) y_i + \lambda p_{\tilde{\theta}}^{x_i})^\top \log p_\theta^{x_i} \tag{13}$$

The Cocktail loss encourages the model to learn to output a mixture of the hard label and the soft label on the training data.

## C.1 Proof of Proposition 1

*Proof.* By induction on $t$. By zero initialization, the statement holds with $g_0 = 0$.

Suppose for iterations up to $t$, we have $\theta_j^{(t)} = g_t$ for all $j \notin \mathcal{S}$. We focus on some $j \notin \mathcal{S}$ WLOG. The gradient for $\theta_j^{(t)}$ is

$$\nabla_{\theta_j^{(t)}} \ell(\theta^{(t)}) = \nabla_{\theta_j^{(t)}} \left( -\sum_{i=1}^{n} y_i^\top \log p_\theta^{x_i} \right) \tag{14}$$

$$= \sum_{i=1}^{n} -y_{ij} x_i + \frac{\exp((\theta_j^{(t)})^\top x_i)}{\sum_{j'} \exp((\theta_{j'}^{(t)})^\top x_i)} x_i \tag{15}$$

$$= \sum_{i=1}^{n} \frac{\exp((\theta_j^{(t)})^\top x_i)}{\sum_{j'} \exp((\theta_{j'}^{(t)})^\top x_i)} x_i \tag{16}$$

$$= \sum_{i=1}^{n} \frac{\exp(g_t^\top x_i)}{\sum_{j'} \exp((\theta_{j'}^{(t)})^\top x_i)} x_i \tag{17}$$

since $y_{ij} = 0$ for all $i$ as $j$ is unseen. This gradient does not vary with $j$, which shows the result. $\square$

## C.2 Proof of Proposition 2

*Proof.* Let $Y_\theta \in \mathbb{R}^{n \times k}$ denote the predicted class probabilities of the model on the training data, and $Y_{\tilde{\theta}}$ be the class probabilities of the prefix-tuned model on training data. With $\lambda = 1$, the gradient of the loss is

$$\nabla_\theta \ell_{ct}(\theta) = \sum_{i=1}^{n} -p_{\tilde{\theta}}^{x_i} x_i^\top + p_\theta^{x_i} x_i^\top \tag{18}$$

$$= -Y_{\tilde{\theta}} X + Y_\theta X \tag{19}$$

Since $\theta$ is the minimizer, this gradient is 0 and thus

$$Y_{\tilde{\theta}} X = Y_\theta X \implies Y_{\tilde{\theta}} = Y_\theta \tag{20}$$

for $X$ with full column rank (right-invertible). We let the right-inverse of $X$ be $X^\dagger$. This implies that for all $i$,

$$\frac{\exp(\tilde{\theta} x_i)}{\sum_j \exp(\tilde{\theta}_j^\top x_i)} = \frac{\exp(\theta x_i)}{\sum_j \exp(\theta_j^\top x_i)}. \tag{21}$$

In matrix form, this is equivalent to

$$\exp(\tilde{\theta}X) \cdot \operatorname{diag}(\mathbf{1}^\top \exp(\tilde{\theta}X))^{-1} = \exp(\theta X) \cdot \operatorname{diag}(\mathbf{1}^\top \exp(\theta X))^{-1} \tag{22}$$

$$\implies \tilde{\theta}X - \mathbf{1}(\log \mathbf{1}^\top \exp(\tilde{\theta}X))^\top = \theta X - \mathbf{1}(\log \mathbf{1}^\top \exp(\theta X))^\top \tag{23}$$

$$\implies \theta = \tilde{\theta} + \mathbf{1}\left[\log \mathbf{1}^\top \exp(\theta X)\log \mathbf{1}^\top \exp(\tilde{\theta}X)\right]X^\dagger. \tag{24}$$

On a new input $x$,

$$\theta x = \tilde{\theta}x + \mathbf{1}\left[\log \mathbf{1}^\top \exp(\theta X)\log \mathbf{1}^\top \exp(\tilde{\theta}X)\right]X^\dagger x \tag{25}$$

$$= \tilde{\theta}x + (\log(\alpha) + \mathbf{1}^\top X^\dagger x)\mathbf{1} \tag{26}$$

by using the assumption on normalizing constants. Therefore the probability of the $j$-th class under the model is

$$(p_\theta^x)_j = \frac{\exp(\theta_j^\top x)}{\sum_{j'}\exp(\theta_{j'}^\top x)} \tag{27}$$

$$= \frac{\exp(\tilde{\theta}_j^\top x)\exp(\log(\alpha) + \mathbf{1}^\top X^\dagger x)}{\sum_{j'}\exp(\tilde{\theta}_{j'}^\top x)\exp(\log(\alpha) + \mathbf{1}^\top X^\dagger x)} \tag{28}$$

$$= \frac{\exp(\tilde{\theta}_j^\top x)}{\sum_{j'}\exp(\tilde{\theta}_{j'}^\top x)} = (p_{\tilde{\theta}}^x)_j \tag{29}$$

$$\square$$

