# OpenReview forum: "Ensembles and Cocktails: Robust Finetuning for Natural Language Generation"
_ICLR.cc/2022/Conference — ICLR 2022 Submitted_

### Official Review · Reviewer_rP9V · 2021-10-19

**Correctness:** 3
**Technical Novelty And Significance:** 3
**Empirical Novelty And Significance:** 3
**Recommendation:** 5
**Confidence:** 4

**Main Review:**

* I am not convinced and even surprised by the authors that "ensembling is one of the methods that we propose as well". I think the ensemble method is existing and NOT proposed by authors, and authors only apply it to the setting lightweight fine-tuning and full fine-tuning.
* "In the experiments we present, we focus on monolingual models, and there isn’t much evidence that prefix-tuning works for translation". The argument is problematic. This paper's title is "Ensembles and Cocktails: Robust Finetuning for Natural Language Generation". Since it is for the general Natural Language Generation (NLG) task, I would like to see these results as MT is one of the most important task in NLG. If the proposed method doesn't work for MT, then I would think the paper overclaims its contribution.
-----------------------------------------------------------------------
Strengths
* The idea of combining adapter-finetuning and full-finetuning to handle both in-domain data and out-of-domain data is interesting and clearly motivated.
* This paper is well written and presents their motivation clearly and finally also provides theoretical analysis on multi-class logistic regression to help understand why it can work.

Weakness
* Their methods only perform comparably to the ensemble method but not consistently and significantly better. Although cocktail fine-tuning only does inference once, it's not free and needs a knowledge distillation process before training, which requires inference over whole-training data using the adapter-finetuning model and can be time-consuming. The authors should talk about this limitation in the paper.
* This paper aims at the natural language generation area and should consider adding experiments on datasets for machine translation (MT).  Adding results for machine translation can make this paper more convincing and solid.

**Summary Of The Paper:**

This paper proposes a simple yet effective method, cocktail fine-tuning, for the natural language generation tasks. Their results show that cocktail fine-tuning can handle both In-domain data and Out-of-domain data effectively by combing adapter-finetuning and full-finetuning through knowledge distillation and overall has comparable performance compared to their ensembles. It also provides theoretical analysis on multi-class logistic regression to explain why it works.

**Summary Of The Review:**

This paper combines adapter-finetuning and full-finetuning to handle both in-domain data and out-of-domain data and the idea is interesting. It's also well written and clearly motivated. Further, it provides a theoretical analysis of multi-class logistic regression to explain why it works. However, I think the authors overclaimed their contribution. Although this paper's title is "Ensembles and Cocktails: Robust Finetuning for Natural Language Generation", it lacks machine translation results, one of the most important tasks in natural language generation. In addition, the authors mention that "ensembling is one of the methods that we propose as well", which I don't agree with. I would recommend declining this paper.

---

> ### Author Response · Authors · 2021-11-21
> **Response to rP9V**
>
> We thank Reviewer rP9V for taking the time to review our work and are glad they find our method to be “simple and effective”. We will now address some of their concerns.
>
> > First, Reviewer rP9V is concerned that our “methods only perform comparably to the ensemble method but not consistently and significantly better”.
>
> We would like to clarify that ensembling is one of the methods that we propose as well, and we do not expect Cocktail finetuning to perform better than it. The benefit of cocktail finetuning is that it requires only a single forward pass at inference time, not that it makes training time any faster.
>
> > rP9V suggests “adding experiments on datasets for Machine Translation”.
>
> In the experiments we present, we focus on monolingual models, and there isn’t much evidence that prefix-tuning works for translation, so we defer these experiments to future work.

---

> > ### Comment · Reviewer_K9aD · 2021-11-21
> > **Re: there isn’t much evidence that prefix-tuning works for translation,**
> >
> > Prefix-tuning is simply a parameter-efficient fine-tuning method which can be applied to any pretrained-models with transformers as backbones, I couldn't think of any reason why it can't be applied to translation.

---

### Official Review · Reviewer_ioou · 2021-10-31

**Correctness:** 2
**Technical Novelty And Significance:** 4
**Empirical Novelty And Significance:** 2
**Recommendation:** 5
**Confidence:** 4

**Main Review:**

The current paper looks perfect up to section 4. It properly discusses different fine-tuning schema as well as the trade-off between OOD performance and ID performance with respect to the use of a fune-tuning scheme. It properly defines the problem (i.e., the trade-off) and proposes a simple yet effective way to overcome it. It tests its method on three different NLG tasks: Table2Text Generation, Summarisation, and QA.

However, starting from section 5, it makes me hard to recommend an acceptance for this paper in the current format. This is because of two major concerns.

One is that the interpretations of experimental results are problematic, weight might make the rest discussions and conclusions do not stand anymore. In section 5, regarding Table 2, the authors say "when evaluating ID, full fine-tuning achieves the stronger results, whereas lightweight fine-tuning achieves only 81% of the skyline on average". This statement is only true "on average" and overlooks the huge differences between different tasks. For example, for ID data of the Data2Text task, full fine-tuning achieves a score of 63.25 while lightweight fine-tuning achieve 63.18. There is no significant difference between the performance on ID of these two schema. Meanwhile, in a similar vein, for OOD samples of the summarisation task, I also found no significant difference between the two schema. The situation described by the author only exists when doing question answering. This suggests that such a trade-off is task-dependent to a large extent and it seems to me that if a task relies on the inputs in a more directed way (e.g., QA is often accomplished by simply copying text from inputs), then the trade-off is more significant and the proposed model is more useful. Whereas, if a task relies on the inputs in a more indirect way (e.g., Data2Text requires the generator to plan what to produce in the first place), then the trade-off is less significant and improvement made by the proposed model is less. Anyway, such a phenomenon should not be overlooked and covered by only saying "on average".

The other is that when evaluating the Data2Text generation, only BLEU is used. However, there has been a bank of work that has proved the BLEU has low validity on NLG tasks (e.g., Reiter, 2018). This makes, at least the results on WebNLG, do not reliable.

Reference:
Reiter, E. (2018). A structured review of the validity of BLEU. Computational Linguistics, 44(3), 393-401.

**Summary Of The Paper:**

The present paper first discusses the trade-off between performance for out-of-domain data and in-domain data with respect to whether the model is fully fine-tuned or lightweight fine-tuned on NLG tasks. Second, it argues that such a trade-off is not necessary if one can make use of both of these two fine-tuning schema in a clever way. To this end, it proposes cocktail fine-tuning, which augments full fine-tuning via distillation from a lightweight model and which achieves equal performance as an ensemble of the two fine-tuning schema. At length, this paper also explains the behavior of the cocktail fine-tuning through a toy model.

**Summary Of The Review:**

I generally like the idea of this paper and the paper is perfect up to the place where the experiments are introduced. Discussions of this paper overlook major phenomena in the results, making the discussions and, probably, the conclusions are, in part, wrong.

---

> ### Author Response · Authors · 2021-11-21
> **Response to ioou**
>
> We thank Reviewer ioou for their valuable review. We appreciate their evaluation of our methods as “clever,”  and their evaluation of our setup and motivation as “perfect.”
>
> > First, reviewer ioou is concerned about our interpretation of our experimental results. They are worried that the result that cocktail finetuning and ensembling improve over the baseline methods holds only “on average.”
>
> We acknowledge that reporting the average differences covers over individual distinctions in the tasks. However, it is the case that there are statistically significant differences (taken over random seeds for training, 0.95 confidence) for tasks other than OpenQA. For ID performance, out of 6 settings (across 3 datasets and 2 lightweight finetuning methods), **full finetuning achieves significantly better ID performance than lightweight finetuning in 5 of the 6)**; only in ID accuracy for prefix tuning in WebNLG is the difference not significant. For OOD performance, out of 6 settings (across 3 datasets and 2 lightweight finetuning methods), **lightweight finetuning achieves significantly better OOD performance than full finetuning in 5 of the 6)**; only in OOD accuracy for adapter tuning in XSUM is the difference not significant. See also our response to Reviewer Sg93, where we show the significant differences between our methods and full and lightweight finetuning.
>
> > Reviewer ioou is also concerned with our use of BLEU, which can have issues in evaluating NLG tasks.
>
> We appreciate the concern; however, the NLG tasks we evaluate on are relatively low-entropy and task-driven, not open-ended or chitchat dialogue; BLEU has fewer issues in these cases (as in machine translation,) where different high-quality outputs tend to have similar word sequences.

---

> > ### Comment · Reviewer_ioou · 2021-11-24
> > **Response**
> >
> > > For ID performance, out of 6 settings (across 3 datasets and 2 lightweight finetuning methods), full finetuning achieves significantly better ID performance than lightweight finetuning in 5 of the 6); only in ID accuracy for prefix tuning in WebNLG is the difference not significant. For OOD performance, out of 6 settings (across 3 datasets and 2 lightweight finetuning methods), lightweight finetuning achieves significantly better OOD performance than full finetuning in 5 of the 6); only in OOD accuracy for adapter tuning in XSUM is the difference not significant.
> >
> > It appears to this result could have a different reading: the trade-off does not necessarily hold in 2 out of 3 tasks. I do think this is worth discussing in the paper, but it is overlooked in both the original version and the revised version.
> >
> > > We appreciate the concern; however, the NLG tasks we evaluate on are relatively low-entropy and task-driven, not open-ended or chitchat dialogue; BLEU has fewer issues in these cases (as in machine translation,) where different high-quality outputs tend to have similar word sequences.
> >
> > I am sorry I cannot agree with you. The authors of WebNLG reported (Shimorina et al., 2018) that the sentence-level BLEU is only moderate co-related with the human judgements. Whereas system-level BLEU does not co-related with human judgement。 Both of them are not informative enough here.
> >
> > ps.  it is unclear which BLEU you were reporting.
> >
> > Reference:
> > Shimorina, A., Gardent, C., Narayan, S., & Perez-Beltrachini, L. (2018). WebNLG challenge: Human evaluation results (Doctoral dissertation, Loria & Inria Grand Est).

---

### Official Review · Reviewer_Sg93 · 2021-11-02

**Correctness:** 3
**Technical Novelty And Significance:** 3
**Empirical Novelty And Significance:** 3
**Recommendation:** 6
**Confidence:** 4

**Main Review:**

* Strengthens:
- This paper addressed interesting research topic and the proposed method is interesting and promising.
- This paper is clear and therefore it is easy to follow and to understand.
- The analyses are interesting and provide insightful information.

* Weaknesses:
- The experimental setup of this paper might be problematic. Two out of the three tasks are generation tasks in which BLEU or ROUGE-2 evaluation scores are difficult to interpret, i.e. it is very difficult to judge the effectiveness of the proposed method.
- The experimental results are mixed, i.e. it is difficult to draw strong conclusions from the results. Furthermore, it is not clear where the results are significant different among systems.
- The definitions of ID and OOD are not well described in this paper and need to be improved. Furthermore, it is not clear how many examples of ID and OOD were used in the datasets.



**Summary Of The Paper:**

This paper presents interesting an idea of combining lightweight fine-tuning and full fine-tuning to achieve the best of both approaches, i.e. perform best on out-of-domain and in-domain data. The authors proposed two approaches: a simple ensemble method and a so-called cocktail fine-tuning that combines two fine-tuning methods in one single model. They evaluated their tasks in three datasets: WebNLG, XSUM and OpenQA and obtained mixed results. The authors also provided good analyses for more insights.

**Summary Of The Review:**

Overall, I rate this paper as marginally above the acceptance threshold, mainly because the idea is interesting and somewhat novel.
However, there are some weaknesses esp. in the experimental setup and results that make this paper a borderline paper.

---

> ### Author Response · Authors · 2021-11-21
> **Response to Sg93**
>
> We thank the reviewer for the valuable feedback. The reviewer appreciated the “interesting idea of combining lightweight fine-tuning and full fine-tuning”, and that the “proposed method is interesting and promising”, and notes that the paper provides “good analyses for more insights”. We respond to some concerns below:
>
> > Sg93 is concerned that “Two out of the three tasks are generation tasks in which BLEU or ROUGE-2 evaluation scores are difficult to interpret, i.e. it is very difficult to judge the effectiveness of the proposed method.”
>
> We believe **in the regime we are working in (e.g., BLEU scores less than 40), the metrics give reliable rankings** of the models. In our paper, **since the datasets we use are similar to [1], we follow their evaluation protocols**.  However, we agree that automatic metrics have their flaws, and that in general we should strive to improve these metrics.
>
> > Sg93 is concerned that “the experimental results are mixed, i.e. it is difficult to draw strong conclusions from the results”.
>
> We thank the reviewer for pointing this out. We believe our results do show that cocktail finetuning combines the benefits of full finetuning and lightweight finetuning. The **cocktail model achieves much better results than the worst of full finetuning or lightweight finetuning, and comparable to the best of the two on most settings**. We consider the 6 settings (3 datasets and ID / OOD accuracy for each). Comparing cocktail finetuning (either prefix or adapter version) to the worst accuracy of its constituent components, cocktail finetuning improves accuracy with statistical significance with respect to a 95% interval over all 6 settings. Comparing cocktail finetuning to the best accuracy of its components, the difference is not significant (i.e., they are comparable) for 4/6 settings for the prefix version and 3/6 settings for the adapter version. We will improve the presentation of the results in the paper. We have updated the presentation of the results table to reflect this.
>
> > Sg93 is unclear about the definitions of ID and OOD” and about “ how many examples of ID and OOD were used in the datasets.”
>
> In general, our **definition for ID and OOD are just that the ID and OOD examples are drawn from different distributions**, which are denoted by $P_{id}$ and $P_{ood}$ (Section 3.1). We do not explicitly depend on assumptions about the relationship between ID and OOD in this paper. The **main leverage we use is that lightweight finetuning methods (prefix tuning and adapter tuning) tend to have good OOD accuracy**, possibly by freezing more pretrained parameters that have been trained on a broad distribution. In our experiments, the concrete ID/OOD splits are by topics. **The appendix (Table 5) gives the number of examples for the ID/OOD splits** in each dataset, and section A.2 in the **appendix details the concrete ID/OOD topical splits for our datasets** (which topics are in which split).
>
> [1] Xiang Lisa Li, Percy Liang. Prefix-Tuning: Optimizing Continuous Prompts for Generation, 2021

---

### Official Review · Reviewer_K9aD · 2021-11-03

**Correctness:** 2
**Technical Novelty And Significance:** 1
**Empirical Novelty And Significance:** Not applicable
**Recommendation:** 3
**Confidence:** 5

**Main Review:**

**Strengths**:

- The writing is generally clear.
- The paper is addressing an important phenomenon that parameter-efficient tuning which has deficient ID performance, while full fine-tuning has worse OOD performance.

**Weakness**:

- First of all, the presented methodology is not well motivated. Parameter-efficient tuning aims to reduce number of parameters that need to save for each task by only fine-tuning a small number of additional parameters. However, the proposed method produces an ensemble model that fine-tunes all parameters, losing the parameter-efficiency. If the goal of this paper is to improve the OOD performance of full fine-tuning, it should compare with methods that improve OOD performance which has a vast amount of related work. And this literature review is also missing from the paper. On the other hand, if the goal of this paper is to improve the ID performance of parameter-efficient tuning, then it should propose a method for improving parameter-efficient tuning methods rather than combine it with full fine-tuning.

- Second, although it's unfair to mention recent work after submission time, there is recent work on parameter-tuning that shows proper improvements over parameter-efficient tuning methods could match the performance of full fine-tuning. This makes the motivation of this paper less meaningful.

- Third, the experiments do not reflect how and why the hyperparameters (length of prefix vectors, bottleneck dimension) of parameter-efficient tuning are selected, which is important for the discussions and conclusions. For example, if with tuned hyperparameters, the ID results of parameter-efficient tuning methods could be close to full fine-tuning while preserving good OOD performance, then the proposed method makes no sense any more.

Another minor flaw is that when prefix-tuning is introduces, it's described as "it prepends a sequence of trainable, task-specific prefix vectors to the input", which actually is not true. Prefix-tuning prepends tunable vectors to the projected key and value vectors instead of the input.

Another question is, the reported ROUGE-2 score on XSUM is quite low (21.2) on ID compared to the original reported result in BART paper (22.27) on the full test set. I would guess the ID performance is even higher or at lease to this number. Why is it?

**Summary Of The Paper:**

This paper proposes an ensemble model between a full fine-tuning model and a parameter-efficient fine-tuning model to improve the out-of-distribution (OOD) performance of a full fine-tuning model. The proposed method is inspired by the observation that full fine-tuning model achieves good in-distribution (ID) performance while parameter-efficient finetuning model achieves better OOD performance. There are two ensembling methods presented in the paper: linear interpolation between the predictions of the two models; and distill from the predictions of a parameter-efficient model with ID training data. Improved OOD performance is observed with this ensemble method.

**Summary Of The Review:**

The motivation of this paper is not well-defined. The goal of this paper is to improve the OOD performance of a full-finetuning model, however, it didn't compare with any method on OOD generalization / domain adaptation. The proposed method also makes parameter-efficient tuning method not attractive any more.

---

> ### Author Response · Authors · 2021-11-21
> **Response to K9aD (part 1)**
>
> We thank the reviewer for the valuable feedback.  Overall, K9aD is concerned about our “best of both worlds” motivation, specifically about the need to improve the ID performance of lightweight finetuning, because K9aD believes lightweight finetuning can match full finetuning in ID settings. We think matching performance might be possible for tasks that don't require much deviation from the original LM (such as classification; only a single token must be generated), but tasks like conditional generation (e.g., summarization, open-QA)  require larger deviation from the original LM, leading to an ID performance gap, as we observe.
>
> > K9aD is concerned that “the proposed method produces an ensemble model that fine-tunes all parameters, losing the parameter-efficiency.”
>
> We agree both our proposed methods (ensembles and cocktail) don’t reduce the number of trainable parameters. However, in our paper, we want to emphasize another advantage of lightweight fine-tuning methods --- it is more robust to distribution shifts, and we aim to exploit the robustness advantage. Our goal is to “improve the OOD performance of full fine-tuning without sacrificing the good ID performance” by exploiting robustness of lightweight methods.
>
> > K9AD says we “should compare with methods that improve OOD performance which has a vast amount of related work”
>
> We appreciate the suggestion; however, our setup doesn’t require extra information about the OOD domains or additional unlabeled data, whereas most of the past work on OOD generalization requires either or both. For example, OOD methods like DANN, self-training, and DIRT-T [4,5,6] either require extra unlabeled data or require knowledge of multiple domains in the training data.. Additionally, these methods are typically designed for classification and generalizing them to generation is nontrivial. Our approach only relies on the helpful inductive bias of lightweight fine-tuning, which provides OOD gains for free, and we are not aware of any related work that would make a fair comparison to our approach. The most robust NLG methods rely on large scale pretraining and finetuning  [1,3] and we are already comparing to them.
>
> > Reviewer K9AD believes that “proper improvements over parameter-efficient tuning methods could match the performance of full fine-tuning.” and allude to some recent work after ICLR submission time.
>
> We appreciate the reviewer’s concern, and we are aware that prefix-tuning demonstrates matching performance to full finetuning on some classification tasks (NLU) in a recent paper [2]. However, we believe this doesn’t generalize to harder tasks like generation, as shown in our experiments. Note that our paper uses the same lightweight method (i.e., prefix-tuning) as used in [2]. Therefore, we think our motivation of achieving the best of both ID and OOD worlds is important for language generation and potentially harder tasks.
>
>
> >  Reviewer K9AD says “the experiments do not reflect how and why the hyperparameters (length of prefix vectors, bottleneck dimension) of parameter-efficient tuning are selected”
>
> As mentioned in Section 4.2 and Table 4 of the appendix, we tune the hyperparameters such as prefix length (for prefix-tuning) and bottleneck dimension (for adapters) based on dev set performance. We roughly follow the suggested hyperparameters in the original prefix-tuning and adapters paper to set our hyperparameter candidate set. Also, we have tried different prefix lengths in our preliminary cocktail experiments (e.g., prefixlength = {20, 40}) and they show the same trend as our reported results (ensembles and cocktails get the best of both ID and OOD performances).
>
> > Reviewer K9AD says “if with tuned hyperparameters, the ID results of parameter-efficient tuning methods could be close to full fine-tuning while preserving good OOD performance”
>
> We don’t think this is true for generation tasks. First, we have done careful hyperparameter tuning, as mentioned above. Second, as preliminary experiments, we have also tried distilling from full finetuning to prefix models. We discover that despite these stronger training signals, there is still a non-trivial ID performance gap. This suggests the lightweight models could have some expressivity issues, which cannot be resolved by hyperparameter tuning (e.g., make prefix length longer, or the bottleneck dim larger). Conditioned on this understanding, our proposed approach is well-motivated and solves an important problem.
>
>
> > Reviewer K9AD says “the reported ROUGE-2 score on XSUM is quite low (21.2) on ID compared to the original reported result in BART paper (22.27) on the full test set.”
>
> In order to simulate the domain shift, we have a different training set (xsum_news), compared to the original XSUM dataset; this is also done in [2]. That’s why we won’t expect the result to match the BART paper’s finetuning result on XSUM.

---

> > ### Author Response · Authors · 2021-11-21
> > **Response to K9aD (part 2)**
> >
> >
> >
> > Re: minor flaws
> > We copy the definition of prefix-tuning from the original paper, where the “prefix” refers to the projected (k,v) pairs for all activation layers.
> >
> > [2] P-Tuning v2: Prompt Tuning Can Be Comparable to Fine-tuning Universally Across Scales and Tasks  https://arxiv.org/abs/2110.07602
> >
> > [3] Using Pre-Training Can Improve Model Robustness and Uncertainty https://arxiv.org/pdf/1901.09960.pdf
> >
> > [4] Domain-Adversarial Training of Neural Networks. Yaroslav Ganin, Evgeniya Ustinova, Hana Ajakan, Pascal Germain, Hugo Larochelle, François Laviolette, Mario Marchand, Victor Lempitsky https://arxiv.org/abs/1505.07818
> >
> > [5] Self-training with Noisy Student improves ImageNet classification. Qizhe Xie, Minh-Thang Luong, Eduard Hovy, Quoc V. Le https://arxiv.org/abs/1911.04252
> >
> > [6] A DIRT-T Approach to Unsupervised Domain Adaptation. Rui Shu,Hung H. Bui, Hirokazu Narui, Stefano Ermon. https://arxiv.org/abs/1802.08735

---

> > > ### Comment · Reviewer_K9aD · 2021-11-21
> > > **Disagree with authors' response**
> > >
> > > I appreciate your response, however, I don't agree with you regarding several key points:
> > >
> > > >  For example, OOD methods like DANN, self-training, and DIRT-T [4,5,6] either require extra unlabeled data or require knowledge of multiple domains in the training data.. Additionally, these methods are typically designed for classification and generalizing them to generation is nontrivial.
> > >
> > > In the examples you give, self-training is definitely easy to apply to sequence generation tasks. Also, there are a couple of meta-learning based methods to better adapt to a new domain by utilizing only a small dev set.
> > >
> > > > Our approach only relies on the helpful inductive bias of lightweight fine-tuning, which provides OOD gains for free, and we are not aware of any related work that would make a fair comparison to our approach. The most robust NLG methods rely on large scale pretraining and finetuning [1,3] and we are already comparing to them.
> > >
> > > Then why is your ensemble method (it performs better than the cocktail method) a fair comparison to pretraining and fine-tuning? Your method needs twice number of parameters of that of full fine-tuning.
> > >
> > > > We appreciate the reviewer’s concern, and we are aware that prefix-tuning demonstrates matching performance to full finetuning on some classification tasks (NLU) in a recent paper [2].  However, we believe this doesn’t generalize to harder tasks like generation, as shown in our experiments. Note that our paper uses the same lightweight method (i.e., prefix-tuning) as used in [2]. Therefore, we think our motivation of achieving the best of both ID and OOD worlds is important for language generation and potentially harder tasks.
> > >
> > > The paper after ICLR submission I referred to is exactly on matching the full-finetuning performance on sequence generation tasks, which proposed several important improvements over existing parameter-efficient tuning methods. For the sake of anonymity and fairness, I won't cite it here. I also don't suggest making such assertive statements without any evidence.

---

> > > > ### Author Response · Authors · 2021-11-26
> > > > **Response to K9aD**
> > > >
> > > > > self-training is definitely easy to apply to sequence generation tasks.
> > > >
> > > > If we had access to unlabeled data, we agree that self-training could be applied. However, in our paper we do not have access to any extra unlabeled data. Self-training on the original training data would be very similar to our distillation method.
> > > >
> > > > > meta-learning based methods to better adapt to a new domain by utilizing only a small dev set
> > > >
> > > > Similarly, we don't have access to labeled data from a test domain and we do not give the models knowledge about the existence of domains. In this paper, we study the general robustness of natural language generation models that do not utilize leverage for OOD generalization from the setting itself.
> > > >
> > > > > why is your ensemble method (it performs better than the cocktail method) a fair comparison to pretraining and fine-tuning
> > > >
> > > > While we include the ensemble method in the table, we view this as a skyline which, as the reviewer pointed out, uses more parameters than the other methods. We do not view the ensemble method as a contribution of the paper, but just use the observation that it does well both ID and OOD as motivation for the cocktail method, which has the same number of parameters as full finetuning.
> > > >
> > > > > The paper after ICLR submission I referred to is exactly on matching the full-finetuning performance on sequence generation tasks, which proposed several important improvements over existing parameter-efficient tuning methods. For the sake of anonymity and fairness, I won't cite it here. I also don't suggest making such assertive statements without any evidence.
> > > >
> > > > We would like to clarify that we only asserted (supported by the evidence from experimental results) that the original prefix-tuning method does not match the ID performance of full finetuning on generation tasks. Our paper provides a way of improving this - if there is another paper which does this through a different methodology, we think these works would complement each other and bring more attention to the deficiencies of previous lightweight tuning methods.

---

### Author Response · Authors · 2021-11-21
**Updated Paper & General Response to Reviewers**

We thank the reviewers all for their valuable feedback. We appreciate that the reviewers valued the simplicity yet promise of our methods and insights. A common concern was our interpretation of results; we believe this is largely due to how we presented our results in Table 2. We’ve updated the formatting of our Table 2 (none of the results have changed) to better show that (1) lightweight finetuning usually significantly outperforms full finetuning in-domain, (2) full finetuning usually significantly outperforms lightweight finetuning out-of-domain, and (3) both ensembling and cocktail finetuning significantly outperforms full finetuning OOD and lightweight finetuning ID in all settings. This is discussed in more detail in our individual reviewer responses. Updates to the paper include the new table 2, and the new results discussions (with updated text colored blue.)

---

### Decision · Program_Chairs · 2022-01-20

**Decision:**

Reject

**Comment:**

This paper presents a method for ensembling light fine-tuning methods and full fine-tuning methods to achieve better performance both in-domain and out-of-domain distributions. As authors agree, similar idea has been explored in the computer vision literature. The reviewers like the overall idea of the paper, but they all had some concerns regarding the experiments. The reviewers provide valuable feedback on how to improve the experiments, potentially running the same idea on more datasets and tasks, provide more analyses and discussions on how to understand the results.